# Bioactive Compounds of *Verbascum sinuatum* L.: Health Benefits and Potential as New Ingredients for Industrial Applications

**DOI:** 10.3390/biom13030427

**Published:** 2023-02-24

**Authors:** Pauline Donn, Paula Barciela, Ana Perez-Vazquez, Lucia Cassani, Jesus Simal-Gandara, Miguel A. Prieto

**Affiliations:** 1Faculty of Science, Nutrition and Bromatology Group, Department of Analytical Chemistry and Food Science, Universidade de Vigo, E32004 Ourense, Spain; 2Centro de Investigação de Montanha (CIMO), Instituto Politécnico de Bragança, Campus de Santa Apolonia, 5300-253 Bragança, Portugal

**Keywords:** *Verbascum sinuatum* L., phenolic compounds, iridoids, extraction techniques, biological properties, nutraceutical ingredient, pharmaceutical applications

## Abstract

*Verbascum sinuatum* (*V. sinuatum*) is a plant belonging to the Scrophulariaceae family that has been used as an ingredient in traditional medicine infusions for the treatment of many diseases. The aerial part of this plant is a source of bioactive compounds, especially polyphenols and iridoids. Moreover, antioxidant activity studies have shown that *V. sinuatum* phenolic and flavonoid composition is higher than those in other plants of the same genus. *V. sinuatum* bioactive compound composition could vary according to the harvesting location, growing conditions of the plants, sample preparation methods, type and concentration of the extraction solvent, and the extraction methods. The obtention of these compounds can be achieved by different extraction techniques, most commonly, maceration, heat assisted extraction, and infusion. Nevertheless, since conventional extraction techniques have several drawbacks such as long times of extraction or use of large amounts of solvents, the use of green extraction techniques is suggested, without affecting the efficiency of the extraction. Moreover, *V. sinuatum* bioactive compounds have several biological activities, such as antioxidant, anticancer, cardiovascular, antimicrobial, antidiabetic, and neuroprotective activities, that may be increased by encapsulation. Since the bioactive compounds extracted from *V. sinuatum* present good potential as functional food ingredients and in the development of drugs or cosmetics, this review gives an approach of the possible incorporation of these compounds in the food and pharmacological industries.

## 1. Introduction

The interest in natural ingredients extracted from traditional plants being applied in the production of therapeutics, functional foods, and cosmetics has drastically increased in recent years. Following the growing consumer demand, several industries are now looking toward developing new products based on natural ingredients to produce safe products with associated beneficial health effects. In the big scope of medicinal plants, the genus *Verbascum* (family Scrophulariaceae) consists of a large group of plants with many applications in traditional medicine. The aerial parts of these plant species are used in the treatment of respiratory problems, such as cough and bronchial congestion, and urinary tract infections. Many biological properties have been attributed to these plants, including antioxidant, anticancer, antihistaminic, analgesic, antispasmodic, antiseptic, and sedative properties, which justify their use in traditional medicine [1]. Among this genus, *Verbascum sinuatum* L. (Phylum Tracheophyta, Class Magnoliopsida, Order Lamiales, Family Scropulaceaea, *V. sinuatum*), also known as mullein, is of huge interest, due to its variety of bioactive properties. *V. sinuatum* is an herbaceous plant of 50 to 200 cm in height, with an erect inflorescence stem, covered with trichomes. Leaves are alternate, entire, and serrated. In winter, a rosette of oblong shortly petiolate leaves grow at the plant base, reaching a length around 30–50 cm. In spring, a leafless stem grows from the base, splitting into other stems (50–100 cm), where the lamp-shaped flowers develop. It presents bracts of 3–8 mm, cordate-deltate, shortly cuspidate. The calix is around 2–4 mm, with lanceolate lobes. Flowers have 4–5 yellow petals with a corolla of 15–30 mm diameter and five stamens [2,3]. This species is a biennial plant, with flowering season during early spring to summer, generally in late May to August [4]. This species is predominantly found in the Mediterranean area, especially in the Turkey–Iraq–Iran region, typically found in waste habitats [5].

Flowers and leaves of *V. sinuatum* have been used for diverse purposes, for example, as an ichotoxic (poison) plant to harvest fish in rivers by the ancient Mediterranean or as an insecticidal agent to kill pests in food products, but primarily, in traditional medicine [6]. The leaves, flowers, and roots of *V. sinuatum* are used in traditional medicine in many countries because of their therapeutic properties in relation to their bioactive compounds content. The flavonoid, iridoid, and phenylethanoid glycoside and phenolic acid content of *V. sinuatum* are potentially responsible of its medicinal properties and valuable health benefits (Figure 1). Many studies on this plant have revealed its antioxidant [7], antimicrobial [8], and antimitotic properties [2]. Thus, this plant presents beneficial properties that would be relevant to explore to evaluate its potential for industrial applications.

For the extraction of the bioactive compounds present in *V. sinuatum*, the most used technique was maceration (a conventional method). This technique has some drawbacks, such as the use of high amounts of solvents, in some cases, not eco-friendly ones, and long extraction times. In recent years, to overcome these drawbacks, there has been a migration toward green extraction techniques. These techniques are considered more eco-friendly, since lower amounts of solvents are required, the extraction time is significantly reduced, and depending on the selected method and conditions, thermosensitive compounds are preserved. However, the extraction conditions must be carefully evaluated, to obtain the best recovery yields. Prior to the selection of an emerging extraction method, it is important to compare their benefits and disadvantages. Therefore, Table 1 presents the advantages and drawbacks of the different green extraction methods that can be applied in the extraction of bioactive compounds present in *V. sinuatum*. The extracted compounds can have different applications with respect to their biological properties. In food industries, they could be employed as preservatives due to their antioxidant properties or as ingredients for new product development, such as functional foods. They can also be applied as natural ingredients in the pharmaceutical industry for the development of drugs with lesser adverse effects and to overcome the increasing problem of resistance of some synthetic antibiotics. In cosmetic industries, the extracted bioactive compounds of *V. sinuatum* could be valuable natural ingredients for the formulation of products. To be sure that the extracted bioactive compounds will finally reach their target after consumption or use without considerable deterioration, they can be preserved by encapsulation. In this sense, two major methods could be employed: microencapsulation and nanoencapsulation techniques [9]. The aim of this study was to evaluate the current findings regarding the bioactive compounds of *V. sinuatum*, the extraction methods that can be used to extract these compounds, and the biological properties of the plant extracts, in order to discuss their potential applications as ingredients in the food, pharmaceutical, and cosmetic industries. In addition, this review may permit identification of current knowledge gaps and stimulate the research in an underexploited species.

## 2. Bioactive Compounds

Bioactive compounds are minor chemical compounds present in plants or animals with beneficial health properties with many potential industrial applications [20]. The aerial part of *V. sinuatum* used in traditional medicine have been analyzed by some authors, who have reported that the main bioactive compounds are polyphenols and iridoids (Table 2).

### 2.1. Phenolic Compounds

Polyphenols are ubiquitous secondary metabolites involved in the protection of plants against different stresses, such as adverse climatic conditions or microbial invasions, in addition to other functions in plant physiology [24]. Polyphenols are highly diversified class of compounds, whose chemical structures are shown in Figure 2. These molecules are divided into four major groups: (1) flavonoids (flavones, flavonols, flavanones, isoflavones, anthocyanidins, chalcones, and catechins); (2) phenolic acids (hydroxybenzoic acids and hydroxycinnamic acids); (3) stilbenes (e.g., resveratrol); and (4) lignans (e.g., secoisolariciresinol). Polyphenols are produced in plants through two main pathways: the acetate and shikimic pathways [24]. Many researchers have assigned many biological properties to polyphenols such as antioxidant, anti-microbial, anti-inflammatory, anti-obesity, anti-diabetic, anti-aging, neuroprotective, and anti-carcinogenic properties [25,26]. For this reason, the extraction and characterization of the different classes of the phenolic compounds from natural plants may lead to the development of new ingredients for industrial applications. Concerning these compounds, three different classes have been identified in *V. sinuatum* [6,20,21,22,23]: (1) flavonoids, such as apigenin, luteolin, naringin, rutin, naringenin, plantagonine, rhamnetin, myricetin, hesperetin, cynaroside, apigetrin, hyperoside, chrysin, and quercetin; (2) phenolic acids, such as caffeic acid, caffeic acid hexoxide, *ρ*-Coumaric acid, quinic acid, ursolic acid, chlorogenic acid, cinnamic acid, and gallic acid; and (3) phenylethanoid glycosides, such as *p*-coumaroyl-6-*O*-rhamnosyl aucubin isomer I and *p*-coumaroyl-6-*O*-rhamnosyl aucubin isomer II.

Many studies have evaluated the polyphenolic profile of the extracts of the aerial part of *V. sinuatum*. For instance, a study evaluated the phenolic profile of the ethanolic extracts (maceration at room temperature for 72 h with ethanol at 70%) of *V. sinuatum* flowers from three different origins in Iran [21]. Spectrophotometric results showed that the total phenolic content of the three samples ranged from 29.03 mg gallic acid equivalents (GAE)/g dry weight (DW) to 54.77 mg GAE/g DW, while the total flavonoid content varied between 13.72 mg rutin equivalents (RE)/g DW and 19.91 mg RE /g DW. Further quantitative analysis, carried out with high-performance liquid chromatography with photodiode-array detection (HPLC–PDA), identified apigenin (5.85–13.82 µg/mg DW), luteolin (3.18–16.98 µg/mg DW), naringin (7.10–12.32 µg/mg DW), rutin (4.90–5.55 µg/mg DW), and caffeic acid (4.98–7.78 µg/mg DW). Other authors have evaluated the total phenolic content (TPC), total flavonoid content (TFC), and the phytochemical profile of the compounds present in the methanolic extracts (maceration at room temperature, 24 h) of flowers of different species of the genus *Verbascum*, including *V. sinuatum* [7]. For TPC and TFC, spectrophotometric methods were employed, and the results showed values of 51.94 mg GAE/g DW and 22.57 mg quercetin equivalents (QE)/g DW, respectively, which were the greatest among the studied species. Main phenolic compounds were identified and quantified by HPLC–PDA coupled with mass spectrophotometry (HPLC–PDA–MS). Thirteen phenolic compounds were identified, the largest being the flavonoid rutin (21.00 mg/g DW) and the phenolic acid *ρ*-Coumaric acid (10.22 mg/g DW). Further phytochemical analysis qualitatively identified other 20 compounds in the methanolic extracts, among them flavonoids (naringenin, plantagonine, luteolin, rhamnetin, myricetin, hesperetin, cynaroside, apigenin, apigetrin, hyperoside, and chrysin) and phenolic acids (quinic acid and ursolic acid) [7]. Other authors have performed chromatographic analysis of the phenolic compounds of *V. sinuatum* leaves, extracted by two different methods: heat assisted extraction with ethanol 80% *v*/*v*, at 40 °C, under stirring at 100 rpm for 1 h (ethanolic extract) and infusion into boiled water for 5 minutes (infusion extract) [23]. The two extracts were lyophilized and redissolved into a solution of 20% ethanol. The chromatographic analysis, achieved through high-performance liquid chromatography coupled with a diode-array detection (HPLC–DAD) system, revealed that the ethanolic extract (EE) presents higher phenolic content than the infusion extract (IE). They identified two phenolic acids: quinic acid at a concentration of 1.25 µg/mL for EE and 1.02 µg/mL for IE and caffeic acid hexoxide at 0.281 µg/mL and 0.112 µg/mL concentration for EE and IE, respectively. They also identified two phenylethanoid glycosides: *p*-coumaroyl-6-*O*-rhamnosyl aucubin isomer I at the concentration of 6.4 µg/mL and 2.7 µg/mL for EE and IE, respectively, and *p*-coumaroyl-6-*O*-rhamnosyl aucubin isomer II at a concentration of 2.3 µg/mL and 1.29 µg/mL for EE and IE, respectively. Their results also revealed that the extraction yield and TPC of the EE estimated at 34.34% and 36 µg/mL, respectively, were higher than those of the IE estimated at 20.29% and 25.2 µg/mL, respectively. Recently, other authors achieved the quantitative analysis of the phenolic compounds present in a methanolic extract (maceration with 80% methanol at room temperature for 24 h) of *V. sinuatum* flowers by HPLC with diode-array detection (HPLC–DAD) [22]. In their findings, the authors detected and quantified six phenolic compounds: quercetin (0.393 g/kg DW), chlorogenic acid (0.154 g/kg DW), cinnamic acid (0.037 g/kg DW), *ρ*-coumaric acid (0.031 g/kg DW), gallic acid (0.015 g/kg DW), and caffeic acid (0.006 g/kg DW). Authors noticed the absence of rutin and apigenin in the analyzed *V. sinuatum* methanolic extract, unlike the previous study [22]. It could be deduced that the polyphenol content of *V. sinuatum* depends on the geographical location and growing conditions as well as methods and parameters applied for their extraction.

### 2.2. Iridoids

Iridoids are cyclopentano[c]pyran monoterpenoids compounds, with a bicyclic **H-5**/**H-9** *β*, *β*-cis-fused cyclopentanopyran ring systema double bond between **C-3** and **C-4**, including a non-substitution at **C-4**. They are a particularly important class of compounds for the *Verbascum* species. The iridoid glycoside content of natural plants such as *V. sinuatum* has been efficient for the treatment of many diseases, such as Alzheimer’s disease, ischemic brain injury, cerebral ischemia, colon cancer, liver injury, diarrhea, food retention, maldigestion, kidney damage, gout, pains, and arrhythmia [27]. Regarding iridoids, the following compounds have been reported by different authors: aucubin, sinuatol, sinuatoside, aucuboside, catalpol, harpagoside, harpagide, pulverulentoside I, catalposide, genipin, catalopl, verbascoside and isoverbascoside, 6-*O*-*β*-D-glucopyranosyl aucubin, 6-*O*-*β*-D-xylopyranosyl aucubin, 6-*O*-*α*-L-sinuatosyl aucubin, and 6-*O*-(3”-*O*-*p*-coumaroyl)-*α*-L rhamnopyranosyl catalpol.

Few studies have evaluated these compounds in *V. sinuatum (*Table 2). A study reported twelve iridoid glycosides compounds: aucubin, 6-*O-β*-D-glucopyranosyl aucubin, sinuatol, 6-*O-β*-D-xylopyranosyl aucubin, 6-*O-α*-L-sinuatosyl aucubin, sinuatoside, aucuboside, catalpol, 6-*O*-(3”-*O-p*-coumaroyl)-*α*-L rhamnopyranosyl catalpol, pulverulentoside I, harpagide, and harpagoside [28]. In a previous study of Selseleh et al., authors quantified an iridoid compound in methanolic extracts of *V. sinuatum*, harpagoside (0.01 mg/g DW). In addition, five other iridoids (catalposide, verbascoside, genipin, catalopl, and aucubin) were qualitatively identified [7]. Further, in the study conducted by Garcia-Oliveira et al., two iridoid glycosides were identified and quantified: verbascoside and isoverbascoside [23]. These authors revealed that the infusion contained a larger amount of verbascoside (13.1 µg/mL) than the ethanolic extract (12.4 µg/mL), while the isoverbascoside content was found to be larger in the extract (2.5 µg/mL) compared to the infusion (2.41 µg/mL). Overall, these findings show that the chemical structures of compounds could affect their affinity with solvents and consequently the extraction yield.

## 3. Extraction Techniques

The screening of the bioactive compounds in the plant extract of *V. sinuatum* has revealed that they are mainly phenolic compounds and iridoids. As previously mentioned, these compounds are widely reported for their beneficial properties. Therefore, obtaining extracts rich in these compounds may be of interest for potential industrial applications. Nevertheless, for an efficient recovery of these compounds (in terms of target compounds, quantity, quality, biological properties, and environmental safety), the methods, type of solvent, and extraction conditions should be carefully considered and optimized [13]. In the following sections, the extraction methods applied to extract *V. sinuatum* found in the literature have been discussed.

### 3.1. Conventional Extraction Techniques

The extraction of bioactive compounds can be done through conventional extraction techniques such as Soxhlet, maceration, and hydro-distillation [12]. These methods use selective solvents that affect the extraction yield of the targeted compounds based on their affinity. The commonly used solvents for these techniques are water, ethanol, methanol, chloroform, dichloromethanol, ether, and acetone [10]. In these methods, for the solvent selection, both polarity and affinity of compounds, environmental issues, toxicity, and cost must be considered. Some authors have explored conventional extractions techniques such as infusion with boiled water, heat assisted extraction (HEA), and maceration into ethanol or methanol to recover bioactive compounds from *V. sinuatum* [7,22]. From a study conducted by previous authors, the extraction yield was higher for HAE using 80% ethanol for 1 h at 40 °C under 100 rpm stirring compared to extraction by infusion into boiled water for 24 h [23]. Thus, the extraction yield of bioactive compounds is affected by the extraction conditions such as solvent (type and concentration), temperature, time, and agitation. Even though there are not many reported data comparing the extraction efficiencies of conventional techniques for *V. sinuatum*, some authors have investigated other *Verbascum* species. For example, a study carried out the extraction of bioactive compounds from *Verbascum bombyciferum* by two methods: homogenizer assisted extraction and maceration with two solvents (water and methanol). For homogenizer assisted extraction, authors obtained extraction yields of 99.750 mg/kg using methanol and 31.018 mg/kg using water. On the other hand, for maceration, the extraction yields were 272.862 mg/kg with methanol and 195.503 mg/kg with water. In the two methods, the extraction yields were higher with methanol as solvent compared to those with water, while the maceration method presented a higher recovery. Nevertheless, they found that the highest TPC and TFC was observed for the homogenizer assisted extraction/methanol (41.80 mg GAE/g and 39.44 mg RE/g), while total bioactive compound extraction was higher when methanol maceration was applied, obtaining 272.862 mg/kg. These authors also showed that, in most assays, the extraction method using methanol as solvent presents the highest antioxidant activity. The authors concluded that methanol was the best extraction solvent and maceration the best conventional extraction method [29]. Another study evaluating the chemical constituents of *Verbascum euphaticum* and *Verbascum oocarpum* applying maceration at room temperature for 24 h using hexane, ethyl acetate, methanol, and methanol: water 80% was also carried out, achieving similar results. For both *Vebascum* species, methanol extracted the highest yields of TPC and TFC. Thus, 50.20 mg GAE/g and 28.47 mg RE/g of TPC and TFC, respectively, were extracted from *Verbascum euphaticum.* The same trial was applied in *Verbascum oocarpum*, obtaining 61.23 mg GAE/g and 49.97 RE/g of TPC and TFC, respectively [30].

### 3.2. Green Extraction Techniques

The emerging green extraction techniques are increasingly preferred for extraction of bioactive compounds. They have the advantage of preserving the environment, speeding up the extraction (lesser time), and enhancing the extraction yields [19]. The major green extraction techniques used in the extraction of bioactive compounds are supercritical fluid extraction (SFE), pulsed electric field (PEF), microwave assisted extraction (MAE), and ultrasound assisted extraction (UAE).

#### 3.2.1. Supercritical Fluid Extraction

Supercritical fluid extraction (SFE) is a novel technology used in the extraction of many natural chemical compounds such as flavonoids, carotenoids, fatty acids, or essential oils [19]. Supercritical state is induced by the temperature and pressure of fluids up to the value of their critical points. In that state, the fluid has both gas-like and liquid-like properties that leads to the reduction of extraction time and an increase in the extraction yields. There are several chemical solvents considered as supercritical fluids, but the most employed is CO_2_ because it possesses low values of critical temperature and pressure and is non-toxic [31,32]. This technique has a reduced processing time, solvent amount, and energy consumption and so is considered eco-friendly. However, it possesses some inconveniences, such as excessive cost of equipment and the fact that the use of CO_2_ as extraction fluid only allows the extraction of non-polar compounds [14].

The use of SFE in the recovery of bioactive compounds from flowers has recently increased according to the reported studies [33]. The SFE of bioactive compounds from flowers such as from *V. sinuatum* could present the same advantages as in reported studies on flowers, according to their reported phenolic compounds [7]. Thus, many authors have performed the SFE of phytochemicals in flower samples. In the study conducted by Pimentel-Moral et al., SFE has been used to optimize the extraction of the total phenolic and organic acids from *Hibiscus sabdariffa* calyces [34]. They obtained the highest extraction yield (113 mg/g) under the following processing conditions: a pressure of 25 Mpa, a temperature of 50 °C, and 16.7% ethanol. Another study showed that there is an increase in the efficiency with supercritical fluid extraction of bioactive compounds from sunflower (*Helianthus annuus* L.) while adding 5% of methanol, dimethyl sulphide (DMSO), or water as a modifier [35]. They also found that the best extraction yield process conditions are 50 Mpa of pressure, a temperature of 50 °C, and 5% of water as a modifier. Although supercritical fluid extraction has not had a reported application in the extraction of bioactive ingredients from *V. sinuatum*, it is obvious to notice that it is an open door that needs to be explored.

#### 3.2.2. Pulse-Electric Field

Pulsed electric field (PEF) is a non-thermal method used in the extraction of bioactive compounds. This technology consists of the application of an electric potential to break the cell membranes and release the cellular content, increasing mass transfer and reducing the time of extraction. In this process, the plant material is placed between two electrodes and is subjected to an energy input of 1 to 20 KJ/kg in an electric field of 0.5–10 kV/cm for a short pulse duration (1 µs−1 ms). It is considered a pre-treatment method used to improve the extraction of bioactive compounds.

Many authors have presented the benefits of using PEF before processing the extraction of compounds. The study by Corrales et al. has shown that anthocyanin mono-glucosides are highly extracted with the application of PEF [36]. The same result has been observed by another study regarding the extraction of anthocyanin and polyphenols [37]. Further, other authors have observed an increase in the extraction yields of phytosterol and isoflavonoids by 32.4% and 20–21%, respectively [38]. The drawback of this technology is the damage produced in the cell membrane when the following conditions are applied—between 500 and 1000 V/cm, for 10-4-10-2 s—due to a minor expansion in the temperature. However, PEF is a highly selective method with low energy cost, low time of extraction, and low quantity of solvent requirement [15]. Moreover, a study has shown that, in this technology, the plant cells are destroyed, allowing an increase in the extraction yield of bioactive compounds from plant tissues. PEF is a non-thermal strengthening method, which can therefore improve the extraction rate of bioactive compounds in flowers [33,39]. Although PEF has fewer applications in the extraction of bioactive ingredients from edible flowers, including *V. sinuatum*, this technology is still very promising.

#### 3.2.3. Microwave-Assisted Extraction

Microwave assisted extraction (MAE) is a green extraction technology in which an electromagnetic field, ranging from 300 MHz to 300 GHz, is applied. In MAE, the electromagnetic energy is converted into heat, which helps to break down the cellular components and improve the transfer of compounds into the solvent [19,40]. The application of the MAE technique has the advantages of using equipment of small size, allowing a rapid heat transfer, and increasing the yield. This method is faster than conventional extraction techniques, and it is considered green due to the use of a reduced amount of solvent [16]. Nevertheless, this extraction method is not suitable for heat sensitive compounds [17].

Several studies have been carried out on the MAE of bioactive compounds from plant extracts, presenting its efficiency and rapidity compared to conventional methods or other green extraction techniques. For example, a study found that by employing MAE, the recovery of flavolignin and silybinin from *Silybum marianum* was higher than conventional techniques [41]. Another study proved that the extraction yields of E- and Z-guggolster-one, cinnamaldehyde, and tannin from many plants were higher in MAE than conventional extraction methods and concluded that MAE is faster and more accessible [42]. Further, other authors have reported that microwave-assisted extraction showed better extraction efficiency than ultrasound assisted extraction in the recovery of bioactive ingredients such as proanthocyanidins from the residue of Kushui rose [33]. The exploration of MAE in the recovery of bioactive compounds from *V. sinuatum* could be a particularly good technique to be applied in non-thermosensitive compounds.

#### 3.2.4. Ultrasound-Assisted Extraction

During ultrasound-assisted extraction (UAE), energy is generated by ultrasound waves between 20 kHz to 100 MHz, causing a cavitation phenomenon through a solvent. This phenomenon creates little bubbles that grow and collapse, inducing energy that finally breaks the components of the cells and improves mass transfer to the solvent [43]. Several factors affect the UAE such as particle size, moisture content, sonification power, frequency, time, pressure, temperature, type of solvent, and distribution of ultrasonic waves [44]. Similar to other green techniques, UAE allows the reduction in the extraction time and the volume of solvent. In addition, it is suitable for the extraction of thermolabile compounds.

UAE of bioactive compounds from plant extracts have been subject to many comparative studies against conventional methods. For example, Zu and colleagues showed that UAE of rosmarinic acid and phenol carboxylic acid is more rapid and efficient compared to those in conventional methods [45]. Further, other research proved that the extraction of quercetin and rutin from *Euonymus alatus* are optimal with UAE [46]. Similarly, another study also found that UAE with ionic solvent from *Catharanthus* spp. of three indole alkaloids—catharanthine, vinblastine, and vindoline—was the most efficient method (extraction time from 0.5 h to 4 h and higher yields) compared to conventional techniques [47]. The study achieved by Herrera and Luque de Castro presented best results in the extraction of some phytochemicals (kaempferol, ellagic acid, naringenin, quercetin, naringin, and rutin) from strawberry with the UAE method [48]. *V. sinuatum*, with the reported identified phytochemicals [7], contains compounds such as naringenin as well that can be better extracted by UAE, which makes this technique a very promising one. Nevertheless, further trials considering all the bioactive compounds present in this plant are needed. Concerning other *Verbascum* species, the application of UAE has been reported in a study related to the UAE of functional polysaccharides from *Verbascum thapsus* [49]. Authors found that in comparison to conventional extraction methods (hot water extraction), UAE under the optimal conditions (60 min, temperature 67.5 °C, ultrasonic power of 371.03 W, and solvent/plant material ratio of 40 *v*/*w*) has a higher extraction yield (5.75%) and a higher antioxidant activity (67.66% of DPPH radical scavenging activity) [49].

## 4. Biological Activities of *V. sinuatum*

Motivated by records of the use of this plant in traditional medicine to alleviate various disorders, some studies have evaluated the biological properties of this plant (Table 3). Most are in vitro analyses, so it is necessary to carry out more specific analysis and in vivo studies to gain an in-depth understanding of the mechanisms of action.

### 4.1. Antioxidant Activity

The metabolic processes that are taking place in the human body involve oxidative reactions responsible for the generation of free radicals. In healthy conditions, endogenous antioxidants are scavenging the oxidation of excessively produced free radicals under the regulation of enzymes to maintain and guarantee equilibrium between pro- and antioxidants for the good functioning of the organism. However, when a disruption in this balance appears with an excess of prooxidant production, it leads to a disturbance in the redox signaling, an incapacity of control, and molecular damage. This phenomenon is known as oxidative stress. During this process, many functional and physiological disorders appear, thereby inducing or favoring the development of many non-communicative diseases such as cancer, obesity, diabetes, and cardiovascular and neurological diseases [52,53,54]. Thus, in that case, the organism needs a sufficient intake of compounds with high antioxidant activity. The antioxidant activity of an extract or compound is the capacity to scavenge or delay the oxidation of a free radical, thus avoiding the formation of harmful compounds that can favor or induce the development of many human disorders. The antioxidant activity of the extracts of *V. sinuatum* have been studied by many authors, as presented in Table 3. For example, Moein and colleagues studied the in vitro antioxidant activity of ethanolic extracts of the aerial part of ten medicinal plants, including *V. sinuatum*. From their results, this plant possessed the highest antioxidant activity in terms of 2,2-diphenyl-1-picrylhydrazyl (DPPH) radical scavenging with a half inhibitory concentration (IC_50_) of 263.52 µg/mL and also in terms of ferric reducing capacity, with a value of 85.08 μg/mL. These results were attributed to the TPC, which was estimated at 8.53 mg/g, the highest among the analyzed plants, corroborating the role of these compounds in the biological properties of the plant [51]. Moreover, the study of Karamian and Ghasemlou (2013) focused on the analysis of antioxidant activity of the methanolic extract of *V. sinuatum*. The evaluation was performed using DPPH, metal chelating, and inhibition of linoleic acid oxidation assays. According to the results, the extracts showed great DPPH scavenging activity, with an IC_50_ of 0.11 mg/mL of extract, which was lower than the control ascorbic acid. Regarding metal chelating activity, the extract obtained a value of 10.88 mg/mL. In the oxidation assay, the extract effectively reduced the oxidation of linoleic acid, with a 51.41% inhibition. However, in these two assays, the results were not comparable with those of the control. As in the previous study, the antioxidant activity was correlated with the high content of phenolic compounds (TPC = 118.2 mg GAE/g DW, TFC = 4.87 QE/g DW) [50]. More recently, a study evaluated the antioxidant activity of three fractions (n-Hexane, ethyl acetate, and water fractions) of the methanolic extract of *V. sinuatum*, among other plants of *Verbascum* genus [7]. For DPPH activity, fractions displayed IC_50_ values of 38.15, 23.56 and 27.37 µg/mL, respectively. For the ferric reducing antioxidant power (FRAP) assay, the results were 1553.29, 2650.92, and 1749.03 equivalents Fe^2+^ µM, respectively. As could be observed, *V. sinuatum* fractions displayed better results in the DPPH activity than in the FRAP assay, which also varied depending on the fraction, suggesting that different compounds are involved in the antioxidant properties. In the study conducted by Garcia-Oliveira et al., the antioxidant activity of an infusion extract (IE) and an ethanolic extract (EE) of *V. sinuatum* were evaluated by the thiobarbituric acid reactive substances (TBARS) [23]. As per their findings, the EE presented a higher antioxidant activity than IE with IC_50_, respectively, evaluated at 4.2 and 17.4 µg/mL. According to Amini et al., amongst the methanolic extracts (maceration with methanol 80% at room temperature for 24 h) of nine species of *Verbascum*, *V. sinuatum* unfortunately presented the lowest values of TFC (2.12 g QE/kg DW), total carotenoid content (1.56 g/kg DW), and *β*-carotene content (1.1 mg/kg DW) [22]. Further, the antioxidant capacity measured by the FRAP assay was also the lowest for *V. sinuatum* (1.12 μmol Fe (II)/g DW.

### 4.2. Antimicrobial Activity

The study of the antimicrobial activity of *V. sinuatum* has been performed by different authors. Thus, the results obtained in different studies are compiled in Table 3. In the literature, one of the first studies to evaluate this property analyzed the effects of methanolic extract of *V. sinuatum* inflorescences against four Gram + bacteria—*Staphylococcus epidermidis*, *S. aureus*, *Enterococcus faecalis*, and *Bacillus subtilis*—and eight Gram — bacteria—*Proteus vulgaris*, *Enterobacter aerogenes*, *Enterobacter cloacae*, *Klebsiella pneumoniae*, *Proteus mirabilis*, *Pseudomonas aeruginosa*, *Salmonella typhi*, and *Citrobacter diversus*. The results of this study showed that the methanolic extract is more effective against Gram + bacteria with minimum inhibitory concentrations (MICs) ranging from 16 µg/mL for *S. epidermidis* to 128 µg/mL for *Bacillus subtilis*. On the other hand, MIC values for Gram — bacteria varied between 64 to 256 µg/mL [1]. A further study evaluated the ethanolic extracts of *V. sinuatum* leaves [8]. Authors focused their research on seven pathogens related to urinary tract infections: *E. faecalis*, *Escherichia coli*, *K. pnemoniae*, *P. aeruginosa*, *P. mirabilis*, and *Candida albicans*. They also compared the antimicrobial activity of the plant extract with six commonly used antibiotics (penicillin, tobramycin, ampicillin, nystatin, ketoconazole, and clotrimazole). The studied extract presented a higher antimicrobial activity against *E. faecalis*, *P. mirabilis*, and *C. albicans* compared to those of the six antibiotics, with inhibition zones of 20.0, 18.0 and 20.0 mm and MICs and minimum bactericidal concentrations (MBCs) of 4.0 (8.0), 8.0 (16.0), and 8.0 (16.0) µg/mL, respectively [8]. Another author evaluated the MIC and MBC of three fractions (n-Hexane: NHF, ethyl acetate: EAF and water fractions: WAF) of the methanolic extract of *V. sinuatum* and nine other *Verbascum* species against *Escherichia coli* (Gram +) and *Staphylococcus aureus* (Gram −) among other *Verbascum* genus [7]. Their analysis revealed that, for the Gram (+) bacteria, MICs of the three fractions of *V. sinuatum* were estimated at 32 mg/mL each, while the MBCs were, respectively, 32, 32, and 32 mg/mL for NHF, EAF, and WAF. Besides that, concerning the Gram (−) bacteria, the three fractions have, respectively, presented the MICs of 8, 2, and 16 mg/mL and MBCs of 16, 8, and 16 mg/mL, showing that the ethyl acetate fraction is more efficient against *Staphylococcus aureus* [7]. Recently, the antimicrobial activity of ethanolic and infusion extracts of *V. sinuatum* leaves against six bacteria (*Staphylococcus aureus*, *Bacillus cereus*, *Micrococcus flavus*, *Listeria monocytogenes*, *Enterobacter cloacae*, and *Salmonella typhimurium*) and five fungi (*Aspergillus fumigatus*, *Aspergillus versicolor*, *Aspergillus niger*, *Penicillium funiculosum*, and *Penicillium aurantiogriseum*) were explored by the previous authors [23]. In this study, the MIC of the methanolic extract ranged from 0.12 to 1 mg/mL and from 0.25 to 1 mg/mL for the infusion extract. Regarding the MBC of the methanolic extract, the results were estimated to be between 0.25 and 2 mg/mL while it was between 0.5 and 2 mg/mL for the infusion extract. In addition, concerning the fungi, the minimum fungicidal concentration (MFC) of the ethanolic extract ranged from 0.5 to 1 mg/mL, whereas for the infusion extract, it presented the same MFC of 1 mg/mL for all the five studied fungi. Thereby, these results revealed that the ethanolic extract of *V. sinuatum* presented better antibacterial properties against the studied microorganisms in comparison to the infusion extract.

Among the bioactive compounds present in *V. sinuatum*, acteoside, a phenylpropanoid glycoside, also known as verbascoside, has been reported to possess antiviral properties. Although no studies have been conducted specifically with extracts or isolated acteoside from *V. sinuatum*, other in vitro and in silico studies have confirmed these properties against different viruses [55,56,57]. Therefore, *V. sinuatum* may also possess anti-viral properties.

### 4.3. Anticancer Activity

The research regarding the use of compounds extracted from natural plants has been increasing in recent decades, and many natural compounds have proven their anticancer effects at different levels. Concerning the extract of *V. sinuatum*, few studies have reported their antitumoral activity. Recently, a study carried out the evaluation of the antimitotic activity of the alkaloid fraction of *V. sinuatum* leaf extract on root cell division of *Allium cepa* L. Through a comparative study between the alkaloid fraction and colchicine (an alkaloid positive control), authors evaluated the mitotic, prophase, and aberration indices. Their results showed that the mitotic index and prophase index of the *V. sinuatum* alkaloid fraction (33.00 and 32.16, respectively, shown in Table 3) were close to the values of the positive alkaloid control (35.00 and 32.28, respectively). Further, *V. sinuatum* alkaloid fraction caused a higher number of chromosomal abnormalities, with an aberration index of 21.49, compared to 5.60 for the positive control [2]. Thus, this research established that the alkaloid fraction of the *V. sinuatum* leaves possessed a mitodepressive effect and could be used in studies based on tumor cells to further evaluate its potential anti-cancer properties. Another author studied the cytotoxic activity of ethanolic and infusion extracts of *V. sinuatum* amongst five traditional medicinal plants on four cancer cell lines: MCF-7 (adenocarcinoma of the breast), NCI-H640 (non-small cell of lung cancer), HeLa (carcinoma of the cervix), and HepG2 (hepatocellular carcinoma) [23]. Between the studied medicinal plants, *V. sinuatum* presented the highest effect against all the studied cancer cell lines with growth inhibitory concentrations (GI_50_) ranging from 101.1 to 172.2 µg/mL in the ethanolic extract and from 59.1 to 92.1 µg/mL in the infusion extract. Although few studies have evaluated this property, acteoside has been reported to possess anticancer activities [58]. According to several studies, this compound exerted pro-oxidative, apoptotic, anti-proliferative, anti-angiogenetic, anti-metastatic, and antitumoral effects against various cancers, which have been attributed to the mediation in different signaling pathways [59,60]. To cite some examples, acteoside have been shown to inhibit the proliferation of breast cancer cells with an IC_50_ of 117 μM [61]. Further, this compound was able to inhibit the progression of glioblastoma cells via a blockade of Wnt/beta-catenin signaling, which led to a reduction in cell viability and tumor growth [62]. In the treatment of prostate cancer, acteoside significantly inhibited cell proliferation, migration, and invasion abilities through downregulation of the TGF-β pathway [63].

### 4.4. Neuroprotective Activity

Amongst the range of neurodegenerative diseases, Alzheimer’s disease (AD) is one of the most reported in clinical trials. It is a degenerative disease mostly occurring at an older age, characterized by memory loss and cognitive impairments [64]. Many authors have studied the neuroprotective effects of verbascoside, a biological compound first isolated in *V. sinuatum*. This compound can inhibit the aggregation of amyloid β (Aβ) [65], reduce the expression of caspase 3 [66] or repair the memory impairment in ICR mice [67], and contribute to the increase of the memory abilities of mice with D-galactose and AlCl3-induced Alzheimer’s disease [68]. The endoplasmic reticulum stress (ER-Stress) is correlated to the progression of Alzheimer’s disease [69]. Some authors have studied the effect of verbascoside on APPswe/PSEN1dE9 transgenic (APP/PS1) mice and amyloid β (Aβ)_1–42_-damaged human glioma (U251) cells [68]. Their findings reveal that, in the brains of APP/PS1 mice, verbascoside significantly inhibits apoptosis, improves the viability of cells, and reduces the accumulation of Amyloid β and the formation of neurofibrillary tangles due to hyperphosphorylated tau protein. Further, for Aβ_1-42_-damaged U251 cells, they also found that verbascoside significantly reduces calcium accumulation, inhibits apoptosis, improves cell viability, reduces the intracellular concentrations of ROS, and improves the morphology of mitochondria and ER. They concluded that due to the inhibitory effect of verbascoside on the branches of the unfolded protein response, by attenuating ER stress and preventing apoptosis, they have significant neuroprotective properties and could be potentially used in the treatment of AD. Additionally, through their correlation with the modulation of neuroinflammation via the NF-κB-p65 pathway, some authors have presented verbascoside as a potential ingredient in the formulation of drugs for the treatment and prevention of AD [70]. In the case of Alzheimer’s disease, previous authors have reported that verbascoside extracted from *V. sinuatum* presented beneficial effects on APPswe/PSEN1dE9 transgenic (APP/PS1) mice and amyloid β (Aβ)_1–42_ exposed glioma cells [68]. They found that verbascoside increases the resistance to endoplasmic reticulum stress, which is involved in the progression of Alzheimer’s disease. Through their findings, verbascoside has been presented as a good ingredient in the development of drugs for the treatment of this neurodegenerative disease.

### 4.5. Cardiovascular Protection

Plants have been used for several years in traditional medicine for the treatment of cardiovascular diseases. For example, in a study conducted by some authors, the potential of *Plantago ovata* husks (source of acteoside compounds) in the prevention of hypertension were presented [71]. In the human cardiovascular system, the renin–angiotensin–aldosterone system (RAAS) has a key role in the regulation of electrolyte balance, blood pressure, and cardiovascular development [72]. The abnormal activation of RAAS automatically leads to hypertension. Indeed, in the kidney, the proteinase known as Renin releases and induces the transformation of Angiotensinogen into inactive Angiotensin I, which will later be converted by the action of angiotensin-converting enzyme (ACE) into Angiotensin II (Ang II), a vasoconstrictive compound. By binding to their receptors of type I, they will cause narrowing of capillaries, regulation of aldosterone, and remodeling of vessels and capillaries, leading to cardiac dysfunction due to the consequent high blood pressure [72]. In the research conducted by Tong and co-workers [73], the in vitro inhibitory activity of acteoside on ACE were experimentally confirmed. They conclude on the antihypertensive effects of an acteoside rich plant, through their impact in the reduction of blood pressure and protection of aorta, heart, and kidney. In addition, many authors have also studied the ACE inhibitory activity (ACEI) of twenty-one phenylethanoid glycosides, including acteoside [74]. The IC_50_ values of the 21 phenylethanoid glycosides on ACEI ranged from 0.53 to 15.035 mM with the IC_50_ of acteoside estimated at 2.22 mM. The presence of hydroxyl groups in phenylethanoids was suggested to have allowed the chelation by Zn^2+^ of the ACE site, concluding that acteoside and the other studied compounds are reliable potential sources of natural anti-hypertensive products. *V. sinuatum* is a source of acteoside and may also show potential in the treatment of cardiovascular diseases.

### 4.6. Antidiabetic Properties

An effective strategy in the management of diabetes is the control of glucose concentration in the blood that can be done by reducing the glucose absorption rate. In the human digestive system, the regulation of the digestion of carbohydrate and glucose absorption is controlled by α-Amylase. In a study conducted with acteoside, authors found that it can induce the inhibition of the activity of α-Amylase with an IC_50_ value of 125.21 mg/mL [75]. Other authors reported that acteoside was effective in the management of diabetic neuropathy, chronic hyperalgesia, and allodynia, which are the most predominant complications in diabetes mellitus [76]. Sodium-dependent glucose cotransporter 1 (SGLT1) plays a significant role in the transport of galactose and glucose into the enterocyte in the small intestine. Monitoring the coupling mechanism of SGLT1/GLUT2 is another means of controlling the concentration of glucose in the blood. A study showed that acteoside can inhibit the sodium dependent glucose cotransporter 1-mediated glucose uptake in intestinal epithelial cells [77]. The release of glucose in the blood is controlled by the pancreatic β-cells through the secretion of insulin. The endoplasmic reticulum stress (ER-Stress) leads to a decrease of the biosynthesis and secretion of insulin by the pancreatic β-cells. This stress could be controlled by bioactive compounds extracted from plants through their antioxidant activities. In this sense, one study demonstrated the protective effects of acteoside against oxidative stress and ER-Stress in the pancreatic β cells by reducing the expression of PERK/eIF2α signal [78]. Therefore, considering that *V. sinuatum* contains a significant amount of acteoside, this plant may be a valuable matrix for further anti-diabetic research.

### 4.7. Toxicity and Herb–Drug Interaction of V. sinuatum

In the literature, few studies have reported the toxicity (in-vitro or in-vivo) of *V. sinuatum* preparations and their potential antagonist interaction effect toward many nuclear receptors. In a previous study [23], the in-vitro cytotoxicity of infusion and ethanolic extracts of *V. sinuatum* against the non-cancer cell line PLP2 was evaluated. According to the results, only the infusion exerted toxic activities against normal cells, with a growth inhibitory concentration 50 (GI_50_) of 223.1 µg/mL (Table 3). Even though we have not found any reported studies regarding the antagonist drug interaction effect of the bioactive compounds of *V. sinuatum*, a previous research [79] reported the possibility of natural ingredients extracted from plants, for example, the case of some flavonoids (like apigenin and luteolin that can also be extracted from *V. sinuatum*), in presenting antagonist effects under specific conditions (high dose) with the aryl hydrocarbon receptor (AhR). They reported that this interaction affected the transcription of the genes responsible for the transport of endogenous and xenobiotic elements involved in diseases management. Further, despite the previously cited beneficial biological benefits of these bioactive ingredients or compounds, some authors have revealed that they can also have health side effects and cause drug interactions due to their capacity to downregulate the transcriptional activities of Cytochrome P450 enzymes and P-glycoprotein through their antagonist effects on various others receptors such as Pregnane Xenobiotic Receptor (PXR) and Constitutive androstane receptor (CAR) [80,81].

## 5. Encapsulation of Bioactive Compounds

Bioactive compounds can possess biological properties, but if after consumption they are not bioavailable, the desired or expected effect will not be achieved. Further, bioactive compounds are generally sensitive to light, oxygen, and temperature. Thus, to preserve the chemical structure and bioactivity of compounds after extraction, different encapsulation techniques can be applied depending on particle size and final use [82]. Therefore, encapsulation is a preservation method for bioactive compounds, that allows to control their stability, water solubility and bioavailability as well as the release rate, with the final aim of reaching a targeted site of action during human consumption or utilization [83]. Two main groups of techniques are used in food and pharmaceutical industries for the encapsulation of bioactive compounds: nanoencapsulation (capsule particle size from 10 to 1000 nm) and microencapsulation (capsule particle size from 3 to 800 μm) [9,83,84].

Although the encapsulation of *V. sinuatum* extracts has not been explored yet, some studies have been performed for other *Verbascum* species and the bioactive compounds identified in *V. sinuatum*, which could suggest that the results would be satisfactory. For example, the quercetin incorporation into a nanoemulsion under optimal conditions have been presented by some authors, as a solution for improving the bioavailability after oral administration [85]. Similarly, the encapsulation of *p*-coumaric acid extracted from *Phyllostachys makinoi* with alginate with chitosan as coated materials permitted to avoid low solubility and poor bioavailability [86]. Further, in the study conducted by others authors [87], the efficiency of nanoencapsulation of the extract of *Verbascum flavidum* have been explored. This research showed that, the nano-formulation of *Apo 4 Extract-Poly Lactide-co-Glycolide* of *Verbascum flavidum* water extract have the potential to inhibit the aggregation of Aβ in relation to their high metal chelator activity (Fe^+2^ chelating activity: 0.413 mg/mL) and encapsulation efficiency (EE: 15%). Their stability was optimal when preserved at −20 °C. Thus, regarding the composition and biological activities of the extracts of *V. sinuatum*, they could be a potential matrix for encapsulation essays to preserve their stability and control their release and bioavailability.

## 6. Potential Applications of *V. sinuatum*

### 6.1. Pharmaceutical Industries

As mentioned before, *V. sinuatum* has been used in different traditional formulations for the treatment of many disorders affecting the human system. Some authors have reported their uses in the formulation of treatments against many diseases, including hemorrhoids, diarrhea, gallstone, liver inflammation, ulcer inflammation, teeth pain, gumboil, hoarse, tonsillitis, cold, cough, asthma, bronchitis, hemoptysis, contusion, broken bones, arthrosis, rheumatism, eczema, exanthema, cysts, zits, wounds, ulcers, burns, chilblain, conjunctivitis, otitis, and helminthiasis [88]. Thus, they have positive effects in the circulatory, digestive, and respiratory system and on the skeleton, skin, and in parasitic infections. Most of these pathologies involve anti-inflammatory processes, which can be justified from a scientific point of view considering the chemical composition of *V. sinuatum* [88]. According to the different reported bioactive compounds and biological properties, this plant could be a promising source of bioactive ingredients for pharmaceutical industries. For example, considering the reported antimicrobial activity of *V. sinuatum*, its extract and bioactive compounds could be a matrix for the development of new antibiotics of natural source. Nowadays, much research has been done on different traditional medicinal plants to determine their real potential in the treatment of diseases caused by bacteria, viruses, and fungi, and to seek alternatives to synthetic antimicrobials and deal with the emergence of antimicrobial resistance. Further, the antioxidant assays conducted with different extracts have shown satisfactory results [7,22,23,50,51]. As it has been widely described, oxidative stress is involved in the development of cardiovascular diseases, cancer, obesity, diabetes, inflammatory diseases, aging, and degenerative diseases [89,90]. Therefore, *V. sinuatum* could be a promising matrix for the development of therapeutics to manage the balance of oxidative stress and help in the management of related diseases. However, more in vivo research models and in clinical trials should be performed, to evaluate the full potential and possible negative effects of this plant. Amongst the reported compounds present in *V. sinuatum*, verbascoside, also known as acteoside, has demonstrated potential to act as a natural alternative in the formulation of treatment of cancers [58], Alzheimer’s disease [68], hypertension [73], and diabetes [75,76,77]. Thus, they are examples of compounds of interest extractable from *V. sinuatum* among many others with a possible application as bioactive ingredients in pharmaceutical industries.

### 6.2. Food Industries

Regarding the phytochemical content and biological properties of *V. sinuatum*, this plant may also have considerable potential in food applications. For example, the bioactive ingredients of this plant can be used as natural food preservatives to prolong the shelf life of food products through their antioxidant properties or used as functional ingredients or nutraceuticals, intended to produce functional foods. Many studies have reported the antioxidant properties of extracts of *V. sinuatum* [6,21,22,49,49,50], which could be considered for the development of preservative agents. This is in line with market trends, which seek a lower use of synthetic antioxidants such as butylated hydroxyl anisole (BHA), butylated hydroxytoluene (BHT), and tert-butylhydroquinone (TBHQ) as they have been related to adverse effects on the health of consumers. Some studies have corroborated the efficacy of the use of natural extracts as preservative agents. For example, Efenberger-Szmechtyk and colleagues showed the potential of using plant extracts rich in polyphenols as natural preservatives and antimicrobial agents in meat industries [91]. Considering the phenolic content of *V. sinuatum*, it could be a very promising application area. On the other hand, *V. sinuatum* could be explored as functional ingredients in the production of functional foods. Several authors have successfully employed natural extracts to enhance the composition and biological properties of food. For example, a study reported that during the processing of kefir, yogurt, and cheese, various bioactive ingredients such as phenolic compounds, can be incorporated in milk to produce functional fermented milk products [92]. Some authors have presented an application of phenolic compounds extracted from olive leaf extracts in biscuits preparations. In their processing, they found that samples formulated with the extract exerted antiglycative capacity against pentosidine and Ne-carboxyethyl-lysine [93]. Thus, the phenolic compounds of *V. sinuatum* may have a similar potential.

In view of the various bioactive compounds extracted from *V. sinuatum*, the potential and the possibility of producing nutraceuticals intended to be used as food preservatives or as functional ingredients in the processing of functional foods is noticeable. However, the use of these bioactive compounds should take into consideration in the delivery method and the different processing steps that they may undergo. Therefore, more research is still necessary before the development of food industrial applications.

## 7. Conclusions

Medicinal plants have been used for years, generation after generation, for the treatment of many diseases but some species have not been fully studied in terms of chemical composition and effective biological activity. Thus, some plants are still underestimated in their overall potential, thereby not sufficiently valued, because of the lack of scientific information. The present review has presented the different bioactive compounds that have been identified in *V. sinuatum*, with an emphasis on the phenolic and iridoids reported compounds. It also presents different methods of extraction, from conventional to green techniques of the previously identified compounds, as a directive for the selection of an eco-friendly method to obtain compounds of high purity and yields. The biological properties of different extracts of this plant and compounds have been presented to reveal their potential beneficial health effects. The current information about compounds and biological properties may suggest that this plant has a considerable potential for applications in food, nutraceutical, and pharmaceutical industries. However, in-vitro, in-vivo, and clinical trial studies for the screening of negative interactions and toxicity of the bioactive compounds intended to serve as ingredients in the pharmaceutical or food industries should be a pre-requisite for avoiding possible antagonistic adverse risks and to guarantee consumer safety.

## Figures and Tables

**Figure 1 biomolecules-13-00427-f001:**
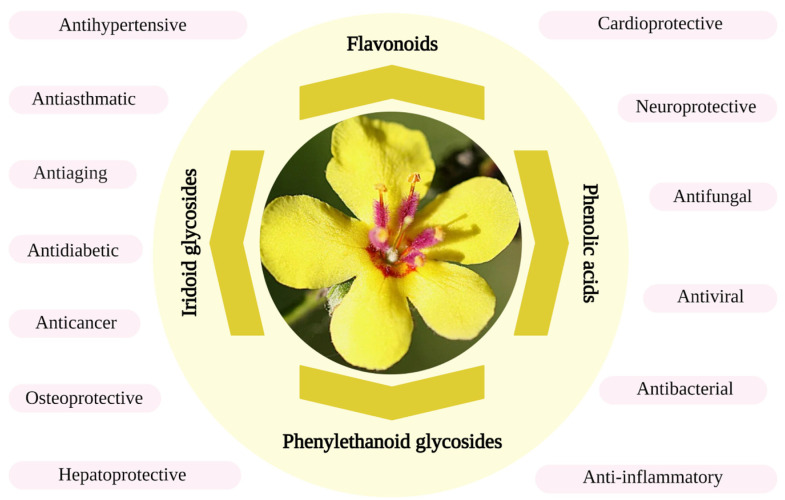
Health benefits of bioactive compounds identified in *Verbascum sinuatum*.

**Figure 2 biomolecules-13-00427-f002:**
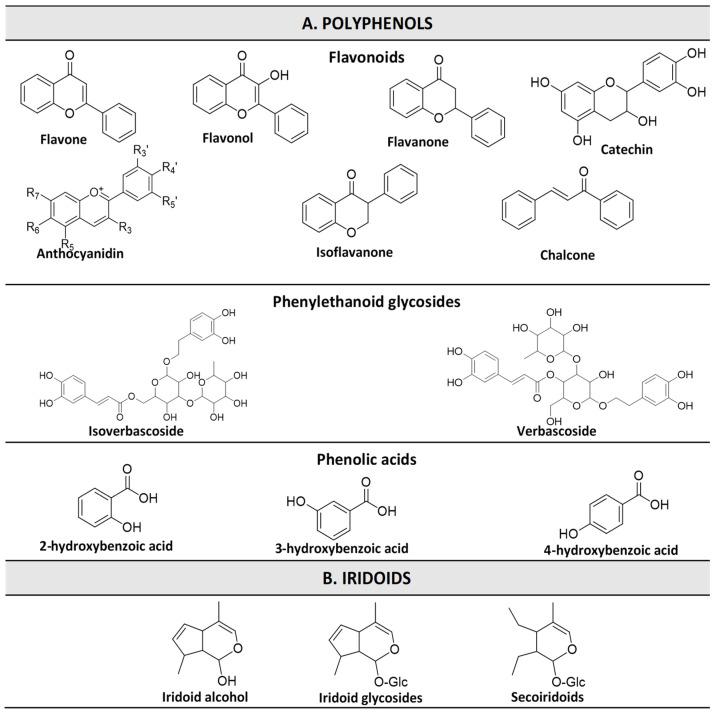
Chemical structures of polyphenols and iridoids present in *V. sinuatum*.

**Table 1 biomolecules-13-00427-t001:** Advantages and drawbacks of conventional and green extraction techniques.

ET	Advantages	Drawbacks	Ref
M ^1^	Requires simple equipment It is an energy saving process Ideal for less soluble compounds needing high prolonged contact Suitable for less potent and cheap drugs	Long ET time up to weeks Low ET yield High quantity of solvent	[10]
SX ^1^	High mass of plant matrix could be used Filtration is not required after ET Simple and repeatable Repeated contact of fresh solvent with the solid matrix	Higher risk of thermal destruction for thermolabile compounds Time consuming and intensive The manipulations of variables are limited Not eco-friendly and requires high quantities of solvents	[11,12]
HD ^1^	Simple and old method Suitable for small-scale industries Provides different options: hydro-distillation, steam and water distillation, direct steam distillation, and hydro-diffusion. The ETs by steam and water distillation are more reproducible	Higher risk of thermal destruction for thermolabile compounds The ET is extremely low and therefore time consuming	[13].
SFE ^2^	Reduced ET time Higher ET yield Less energy consumption Mild temperature and pressure Small volume of solvent Suitable for ET of volatile compounds	Excessive cost of equipment Use of CO_2_ as ET fluid only allows the ET of non-polar compounds	[14]
PEF ^2^	Highly selective Low quantity of solvent and energy cost Rapid ET process	Damage of cell membrane	[15].
MAE ^2^	Small size equipment Allows a rapid heat transfer Increase in the ET yield Reduced amount of solvent	Induces damage of heat sensitive compounds Low drying rate	[16,17]
UAE ^2^	High ET yields Thermolabile compounds are safe ET time is reduced Has simple and small equipment Is more selective and diverse in ET	Solvent is not renewed The additional filtration step can lead to loss or contamination	[18,19]

Abbreviations: ^1^ refers to conventional extraction techniques; ^2^ refers to green extraction techniques; ET: extraction technique; M: maceration; HD: hydro-distillation; SFE: supercritical fluid extraction^2^; PEF: pulse electric field; MAE: microwave assisted extraction; UAE: ultrasound assisted extraction; SX: soxhlet.

**Table 2 biomolecules-13-00427-t002:** Studies on the bioactive composition of *V. sinuatum* extracts.

EX	Part	Cond	Results		Ref.
Maceration (ethanol)	Flower	70%, RT, 72 h	SP (TPC; mg GAE/g DW; TFC: mg RE/g DW)	HPLC–PDA (µg/mg DW)	[21]
			TPC (29.03–54.77), TFC (13.72–19.91)	*Flavonoids*: Apigenin (5.85–13.82); Luteolin (3.18–16.98); Naringin (7.10–12.3); Rutin (4.90–5.55)	
				*PAs*: CAF (4.98–7.78)	
Methanolic (extract)	Flower	RT, 24 h	SP (TPC; mg GAE/g DW; TFC: mg RE/g DW)	HPLC–PDA–MS (mg/g DW)	[7]
			TPC (51.94), TFC (22.57)	*Flavonoids*: Rutin (21); Quercetin (1.51)	
				*PAs*: PCA (0.12); GA (0.09); SAc (2.45); PMCA (10.22); FA (0.45); RA (3.63)	
				*Iridoids*: Harpagoside (0.01)	
Maceration	Flower	80%, RT, 24 h	HPLC–DAD (g/kg DW)		[22]
			*Flavonoids*: Quercetin (0.393)		
			*PAs*: CGA (0.154); CAN (0.037); PMCA (0.031); GAA (0.015); CAF (0.006)		
HAE (ethanol)	Leaf	80%, 40 °C, 100 rmp/1h	SP/ GRV (TPC: µg/mL)	HPLC–DAD (µg/mL)	[23]
			TPC (36), EY (34.34%)	*PAs*: QA (1.25); CAFh (0.281)	
				*PGs*: PMCI (6.4); PMCII (2.3)	
				*IGs*: VB (12.4); IVB (2.5)	
Infusion	Leaf	Water, 5 min	SP/ GRV (TPC: µg/mL)	HPLC–DAD (µg/mL)	
			TPC (25.2), EY (20.29%)	*PAs*: QA (1.02); CAFh (0.112)	
				*PGs*: PCMI (2.7); PCMII (1.29)	
				*IGs*: VB (13.1); IVB (2.41)	

Abbreviations: EX: extraction; COND: condition; IGs: iridoid glycosides; PGs: phenylethanoid glycosides; PAs: phenolic acids; CAF: caffeic acid; CAFh: caffeic acid hexoxide; PCA: protocatechuic acid; GA: gentisic acid; Sac: salicylic acid; PMCA: *p*-Coumaric acid; FA: ferulic acid; RA: rosmarinic acid; GAA: gallic acid; CGA: chlorogenic acid; CAN: cinnamic acid; QA: quinic acid; VB: verbascoside; IBV: isoverbascoside; PMCI: *p*-Coumaroyl-6-*O*-rhamnosyl aucubin isomer I; PCMII: *p*-Coumaroyl-6-*O*-rhamnosyl aucubin isomer II; EY: extraction yield; SP: spectrophotometry; GRV: gravimetry.

**Table 3 biomolecules-13-00427-t003:** Biological activities of *V. sinuatum* extracts.

Extracts	Effects	Ref.
**Antioxidant Activity**		
Methanolic	TPC (118.2 GAE/g DW); TF (4.87 mg QE/g DW); DPPH (IC_50_ 0.11 mg/mL); Fe-Chelate concentration (10.88 mg/mL)	[50]
Methanolic	TPC (51.94 mg GAE/g DW); TF (22.57 mg QE/g DW); DPPH (NHF 38.15, EAF 23.56, WAF 27.37) IC50; µg/mL; FRAP (NHF 1553.29, EAF 2650.92, WAF 1749.03) equivalents Fe^2+^ µM	[7]
Methanolic	TF (2.12 g QE/kg DW); TCC (1.56 g/kg DW); β-carotene content (1.1 mg/kg DW); FRAP (1.12 μmol Fe (II)/g DW)	[22]
Ethanolic	TPC (8.53 mg/g); DPPH (IC50 = 263.52 µg/mL); FRAP (85.08 μg/mL)	[51]
Ethanolic	TBARS (IC50: 4.2 µg/mL)	[23]
Infusion	TBARS (IC50: 17.4 µg/mL)	
**Antimicrobial Activity**		
Methanolic	MIC µg/mL: *S. epidermidis* (16); *S. aureus* (32), *E. faecalis* (32); *P. vulgaris* (64); *P. mirabilis* (64); *C. diversus* (64); *B. subtilis* (128); *E. cloacae* (128); *P. aeruginosa* (128); *S. typhi* (128); *E. aerogenes* (256); *K. pneumoniae* (256)	[1]
Methanolic	MIC mg/mL: *E. coli* (NHF 32; EAF 32; WAF: 32); *S. aureus* (NHF 8, EAF 2; WAF 16)	[7]
	MBC mg/mL: *E. coli* (NHF 32; EAF 32; WAF: 32); *S. aureus* (NHF 16, EAF 8, WAF 16)	
Ethanolic	IZ mm: *E. faecalis* (22); *E. coli* (11); *K. pneumoniae* (12); *P. aeruginosa* (12); *P. mirabili* (18); *C. albicans* (20)	[8]
	MIC µg/mL: *E. faecalis* (4); *E. coli* (500); *K. pneumoniae* (250); *P. aeruginosa* (250); *P. mirabili* (8); *C. albicans* (8)	
	MBC µg/mL: *E. faecalis* (8); *E. coli* (1000); *K. pneumoniae* (500); *P. aeruginosa* (250); *P. mirabili* (16); *C. albicans* (16)	
Ethanolic	MIC mg/mL: *S. aureus* (0.25); *B. cereus* (0.5); *M. flavus* (1); *L. monocytogenes* (1); *E. cloacae* (1); *S typhi* (1); A. *fumigatus* (0.25); *A. versicolor* (0.25); *A. niger* (0.5); *P. funiculosum* (0.12); *P. aurantiogriseum* (0.5)	[23]
	MBC mg/mL: *S. aureus* (0.25); *B. cereus* (0.5); *M.* flavus (2); *L. monocytogenes* (2); *E. cloacae* (2); *S typhi* (2)	
	MFC mg/mL: A. *fumigatus* (0.5); *A. versicolor* (0.5); *A. niger* (1); *P. funiculosum* (0.5); *P. aurantiogriseum* (1)	
Infusion	MIC mg/mL: *S. aureus* (0.25); *B. cereus* (1); *M. flavus* (1); *L. monocytogenes* (1); *E. cloacae* (0.5); *S. typhi* (1); *A. fumigatus* (0.5); *A. versicolor* (0.5); *A. niger* (0.5); *P. funiculosum* (0.5); *P.* aurantiogriseum (0.5)	[23]
	MBC mg/mL: *S. aureus* (0.5); *B. cereus* (1); *M. flavus* (2); *L. monocytogenes* (1); *E. cloacae* (1); *S. typhi* (2)	
	MFC mg/mL: *A. fumigatus* (1); *A. versicolor* (1); *A. niger* (1); *P. funiculosum* (1); *P. aurantiogriseum* (1)	
**Anticancer Activity**		
Alkaloid fraction	Mitotic index (33.00); prophase index (32.16); aberration index (21.49)	[2]
Ethanolic	GI50 µg/mL: MCF-7 (125.9); NCI-H640 (140); HeLa (101.1); HepG2 (172.2); PLP2 (>400)	[23]
Infusion	GI50 µg/mL: MCF-7 (74.8); NCI-H640 (92.1); HeLa (59.1); HepG2 (65.4); PLP2 (223.1)	[23]

Abbreviations: DPPH: 2,2-Diphenyl-1-picrylhydrazyl radical scavenging; FRAP: ferric reducing antioxidant power; TBARS: thiobarbituric acid reactive substances; IC_50_: half inhibitory concentration; NHF: n-Hexane fraction; EAF: ethyl acetate fraction; WAF: water fraction; MIC: minimum inhibitory concentration; MBC: minimum bactericidal concentration; MFC: minimum fungicidal concentration; GI_50_: growth inhibitory (50%) concentration; MCF-7: adenocarcinoma of the breast; NCI-H640: non-small cell of lung cancer; HeLa: carcinoma of the cervix; HepG2: hepatocellular carcinoma; PLP2: non-tumor cell line obtained from pig liver; TPC: total phenolic content; FRAP: ferric reducing capacity; TF: total flavonoid; TCC: total carotenoid content; *S. epidermidis*: *Staphylococcus epidermidis*; *S aureus*: *Staphylococcus aureus*; *E. faecalis*: *Enterococcus faecalis*; *P. vulgaris*: *Proteus vulgaris*; *P. mirabilis*: *Proteus mirabilis*; *C. diversus*: *Citrobacter diversus*; *B. subtilis*: *Bacillus subtilis*; *E. cloacae*: *Enterobacter cloacae*: *P. aeruginosa*: *Pseudomonas aeruginosa*; *S. typhi*: *Salmonella typhi*; *E. aerogenes*: *Enterobacter aerogenes*; *K. pneumoniae*: *Klebsiella pneumoniae*; IZ: inhibition zones; *E. coli*: *Escherichia coli*; *C. albicans*: *Candida albicans*; *M. flavus*: *Micrococcus flavus*; *L. monocytogenes*; *Listeria monocytogenes*; *A. fumigatus*: *Aspergillus fumigatus*; *A. versicolor*: *Aspergillus versicolor*; *A. niger*: *Aspergillus niger*; *P. funiculosum*: *Penicillium funiculosum*; *P. aurantiogriseum*: *Penicillium aurantiogriseum.*

## Data Availability

Not applicable.

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
