# Peer review of "Bioactive Compounds of Verbascum sinuatum L.: Health Benefits and Potential as New Ingredients for Industrial Applications"

_biomolecules, 2023, doi:10.3390/biom13030427_

Round 1
Reviewer 1 Report
Journal- Biomolecules
Manuscript ID: biomolecules-2168109
Title-"Bioactive compounds of Verbascum sinuatum L.: health benefits and potential as a new ingredient for functional foods and pharmaceutical industries"
Reviewer comments
1. The morphological characteristics and distribution are essential for phytochemists, taxonomists and persons working on the medicinal properties of herbs. The authors have defined these points in the introduction section of the present article. However, it would be good if the authors could use a separate heading, “Plant morphology, classification and geographical distribution” for this section.
2. The quality of Figure 1 is very inferior. The authors need to redraw it.
3. Throughout the manuscript, sometimes authors have used Verbascum sinuatum and randomly V. sinuatum. Technically, this is incorrect. As per the binomial nomenclature, in any scientific documentation, first mention the name of the organism in full form (e.g., Verbascum sinuatum), and later, it can be written in short (e.g., V. sinuatum). Authors need to follow the rule of this classification system and maintain uniformity throughout the manuscript. Besides, it will be excellent if authors first read about “binomial nomenclature”.
4. In heading 2, “Bioactive compounds”, the authors first need to mention what phytochemicals were reported from V. sinuatum, then, in a separate heading, they can describe their isolation. In the “Bioactive compounds” section, they described both phytochemicals and isolation together without maintaining any continuity, which looks abrupt and messy. The authors need to improve this section.
5. The quality of figure 2, “Chemical structure ……V. sinuatum” is inferior. At first sight, it appears that the authors have just copied and paste. Therefore, the authors must redraw this figure in ChemDraw.
6. What is the antioxidant activity? What is the role of antioxidant activity in reducing the pathogenicity of disease? In short first describes these points, then explain the antioxidant property of V. sinuatum.
7. Table 1, should contain antioxidant activity, not all the biological activities.
8. What is the in-vitro and in-vivo toxicity level of V. sinuatum. The authors need to describe it adequately.
9. Plants, including V. sinuatum is used as essential ingredients in the preparation of herbal medicine, dietary supplements, and functional foods, etc. Like other plants, the phytochemicals of V. sinuatum also act as agonists or antagonists with various nuclear receptors such as PXR, CAR and AhR and regulate their downstream genes, such as CYP3A4, CYP2C9 and P-gp transporters and may induce herb-drug interaction. This is an important and emerging field of phytomedicine. Authors can add a separate section on “Herb-drug Interaction of V. sinuatum” and for the basic introduction of this section following articles https://doi.org/10.1016/j.jep.2022.115822; https://doi.org/10.1159/000334488; https://doi.org/10.1016/j.phymed.2020.153416; https://doi.org/10.1002/fft2.110; https://doi.org/10.1016/j.heliyon.2020.e05357; https://doi.org/10.1016/j.jep.2022.115822; https://doi.org/10.3390/jcm11061567; doi: 10.3390/molecules26082315 etc. should be helpful and can be used as references.
10. The authors did not mention functional foods, dietary supplements and pharmaceutics prepared from V. sinuatum. Then, why did the authors used the term “functional foods and pharmaceutical industries” in the title? To justify it, authors need to add commercially available preparation of functional foods, dietary supplements and pharmaceutics prepared from V. sinuatum.
11. Authors should at least properly format the manuscript before submission.
Overall the present form of the review article is inferior and needs improvement at various levels.
Author Response
Dear Reviewer,
Thank you very much for your comments and suggestions. Please find in the attachment a detailed response to your considerations. Note that in black color are your questions, in red color our answers, and in blue color are the changes in the manuscript. Also, we have attached the manuscript with all the changes performed using the Track & Changes tool from Microsoft.
Yours sincerely,
The authors

Reviewer 2 Report
The article clearly and condensed presents the current state of knowledge about Verbascum sinuatum L. as a potential raw material for the production of nutraceuticals. The topic is innovative, the manuscript is structured correctly, the language is appropriate. The authors used as many as 84 literature items.
I recommend for publication in Biomolecules after a major revision - comments in the attached file.

Author Response

(The authors gave the same response as above.)

Reviewer 3 Report
In the review entitle “Bioactive compounds of Verbascum sinuatum L.: health benefits
and potential as a new ingredient for functional foods and pharmaceutical industries”, the authors revised the available literature in this species considering recent publications.
In general, the manuscript is well written including figures and tables.
Some minor comments and suggestions are listed below:
Figure 1 have low resolution, increase it.
Line 341: correct position of the subtitle 4.1. Antioxidant activity.
Line 545: revise it particle size in nano and microencapsulation.
Line 600: “Many studies”, include those references.
Author Response

(The authors gave the same response as above.)

Round 2
Reviewer 1 Report
Journal- Biomolecules (ISSN 2218-273X)
Manuscript ID: biomolecules-2168109
Title-"Bioactive compounds of Verbascum sinuatum L.: health benefits and potential as a new ingredient for functional foods and pharmaceutical industries"
Reviewer comments
Dear Editor,
The authors have fulfilled all the queries/comments as it was asked previously. Now the manuscript is well written. I believe that it is a nice piece of work for being published in the Biomolecules. Finally, I recommend that the paper be accepted for publication in its present form.
Decision- Accept
Reviewer 2 Report
Accept in present form